# The N-Terminal Mutations of cMyBP-C Affect Calcium Regulation, Kinetics, and Force of Muscle Contraction

**DOI:** 10.3390/ijms252413405

**Published:** 2024-12-13

**Authors:** Salavat R. Nabiev, Galina V. Kopylova, Victoria V. Nefedova, Alexander M. Matyushenko, Daniil V. Shchepkin, Sergey Y. Bershitsky

**Affiliations:** 1Institute of Immunology and Physiology, Russian Academy of Sciences, 620049 Yekaterinburg, Russia; salavatik2003@gmail.com (S.R.N.); g_rodionova@mail.ru (G.V.K.); cmybp@mail.ru (D.V.S.); 2Research Center of Biotechnology, Russian Academy of Sciences, 119071 Moscow, Russia; victoria.v.nefedova@mail.ru (V.V.N.); ammatyushenko@mail.ru (A.M.M.)

**Keywords:** cardiac myosin binding protein C, hypertrophic cardiomyopathy mutations, actin-myosin interaction, slow skeletal muscle fiber, fiber force kinetics, fiber stiffness, tension recovery

## Abstract

The cardiac myosin binding protein-C (cMyBP-C) regulates cross-bridge formation and controls the duration of systole and diastole at the whole heart level. As known, mutations in cMyBP-C increase the cross-bridge number and rate of their cycling, hypercontractility, and myocardial hypertrophy. We investigated the effects of the mutations D75N and P161S of cMyBP-C related to hypertrophic cardiomyopathy on the mechanism of force generation in isolated slow skeletal muscle fibers. The mutation D75N slowed the kinetics of force development but did not affect the relaxation rate. The mutation P161S slowed both the relaxation and force development. The mutation D75N increased the calcium sensitivity of force, and the mutation P161S decreased it. The mutation D75N decreased the maximal isometric tension and increased the tension and stiffness at low calcium. Both mutations studied disrupt the calcium regulation of contractile force and affect the kinetics of its development and thus may impair cardiac diastolic function and cause myocardial hypertrophy.

## 1. Introduction

Hereditary hypertrophic cardiomyopathy (HCM) caused by the disruption of actin-myosin interactions results in accelerated cross-bridge cycling mainly due to mutations in sarcomeric proteins. The most frequently mutating genes are *MYH7* and *MYBPC3*, which encode myosin beta-chain and cardiac myosin binding protein-C (cMyBP-C), respectively [1,2,3]. cMyBP-C is essential for maintaining the sarcomere structure and coordinates the mutual sliding of thick and thin filaments during myocardial contraction [4,5,6,7]. The absence of cMyBP-C in mouse models demonstrates significant changes in sarcomere structure, contractile function, cardiac hypertrophy, and severe heart failure, indicating the importance of cMyBP-C for the base cardiac functions [8,9,10].

The cMyBP-C molecule consists of 11 domains conventionally divided into three parts: N-terminal (C0-C2), central (C3-C6), and C-terminal (C7-C11). Each of them plays a specific role in regulating actin-myosin interactions and maintaining the structure of the cardiomyocyte sarcomere. The N-terminal domains contain binding sites for many sarcomeric proteins and affect the contractile function of the myocardium. The N-terminal part interacts with actin, myosin subfragments S1 and S2, and tropomyosin [11,12]. It has been shown that the N-terminal fragments of cMyBP-C increase the calcium sensitivity of tension in permeabilized myocytes [13] and skeletal muscle fibers [14] and accelerate tension recovery in skinned rat trabeculae [15].

Point mutations are found along the entire length of the cMyBP-C molecule. The mechanisms of HCM occurrence caused by point mutations remain poorly understood, in contrast to the truncated forms of cMyBP-C, which lead to a decrease in the amount of protein in the sarcomere, acceleration of cross-bridge cycling, an increase in force, and as a consequence, myocardial hypertrophy [16]. Despite the importance of the N-terminal domains in myocardial contraction, by now, the action mechanism of only a few point mutations located in this part has been studied: E258K and Y235S in the C1 domain, E334K and L352P in the m-motif [17,18,19,20]. Experiments on isolated myocardial preparations, iPS cells, and at the molecular level have shown that these mutations affect calcium dynamics in the cytosol and the rate of cross-bridge cycling [17,18,19,20]. Mutations in the C0 domain are poorly understood.

In recent work, we investigated the mutations D75N and P161S in the C0 and C1 domains at the molecular level using an in vitro motility assay [21]. We showed that both mutations reduced the calcium sensitivity of the sliding velocity of thin filaments and impaired their activation. In addition, the mutation D75N alters nucleotide exchange kinetics, and molecular dynamics simulations indicate that the mutation D75N affects myosin S1 function.

However, the in vitro motility assay does not allow studying the effect of cMyBP-C and its mutations on the mechanism of force generation, the kinetics of force development, and the fraction of attached cross-bridges. For this reason, we used permeabilized slow skeletal muscle fibers from rabbits as a well-structured experimental object, the myosin heavy chains of which are similar to those of cardiac muscle. Unlike isolated cardiomyocytes, skinned skeletal muscle fiber is convenient for studying calcium regulation and contraction kinetics [14,22]. This work aimed to study the effect of two HCM-related mutations of cMyBP-C, D75N and P161S, on the mechanical and kinetic characteristics of the contraction of a single slow skeletal fiber. Our results could help to explain the effects of mutations on the properties of cross-bridges and the contractile function of striated muscles.

## 2. Results

The effects of mutations P161S and D75N in the N-terminal C0-C2 fragments of cMyBP-C on the mechanical and kinetic characteristics of isolated permeabilized fibers from slow skeletal muscle (*m. soleus*) of rabbits were studied. First, we investigated the effect of cMyBP-C fragments on the calcium dependence of muscle fiber tension and stiffness. The results were obtained by averaging the calcium curves for eight fibers for each cMyBP-C fragment studied. Only those fibers that developed more than 100 kPa at full activation at 15 °C were analyzed. The normalized *p*Ca-tension dependences of slow fibers with C0-C2 fragments are shown in Figure 1a.

The maximum tension at a saturated calcium concentration (*T*_max_) developed by muscle fibers in the control, i.e., without added cMyBP-C fragments, as well as the minimum tension (*T*_min_) in the absence of calcium, varied in a wide range: *T*_max_ from 106 kPa to 248 kPa and *T*_min_ from 1 kPa to 11 kPa. For this reason, we normalized the tension by adding cMyBP-C fragments (*T*_maxC_ and *T*_minC_) to the corresponding values in the control for each fiber, after which the ratios were averaged. The stiffness values (*S*_max_, *S*_maxC_, *S*_min_, *S*_minC_), which closely correlated with the fiber tension, were also processed. The range of fiber stiffness values in the control was from 10 MPa to 24 MPa (S_max_) at a saturating concentration of calcium (*p*Ca 5.5) and from 0.7 MPa to 2.3 MPa (S_min_) in the absence of calcium (*p*Ca 9). The kinetics of tension recovery after a rapid shortening of the fiber by ~10% and returning to the initial length (release-stretch protocol [23], Figure 2) changed approximately equally and independently of the maximum tension developed by each fiber, and so the rate constants of tension redevelopment *k*_trC_ and *k*_tr_ (with and without C0-C2 fragments, respectively) were averaged over all fibers and then normalized. The average value of *k*_tr_ in the control was 3.55 ± 0.07 s^−1^. The *k*_tr_ values are in the range of earlier published data [24,25].

Table 1 shows the average values of the parameters of the calcium dependence of the fibers tension: calcium sensitivity, *p*Ca_50_, the Hill coefficient, *h*; and also the ratios of the maximum and minimum tension values, *T*_maxC_/*T*_max_ and *T*_minC_/*T*_min_.

The addition of the 5 µM WT C0-C2 fragment increased the Ca^2+^ sensitivity of tension relative to the control, shifting *p*Ca_50_ to the left by 0.3, and decreased the slope of the calcium dependence of tension, i.e., the Hill coefficient (Figure 1a, Table 1). The D75N C0-C2 fragment at the same concentration increased the calcium sensitivity of tension to a greater extent than the WT fragment. It significantly decreased the slope of the calcium curve compared to both the control curve and that with the WT C0-C2 fragment (Figure 1a, Table 1). On the contrary, the P161S C0-C2 fragment decreased the calcium sensitivity relative to both the control and the WT C0-C2 fragment (Table 1) and did not affect the slope regarding the control and increased it relative to the WT fragment (Figure 1a, Table 1).

As seen in Table 1, WT C0-C2 and P161S C0-C2 did not affect the maximum and minimum fiber tension. Only the D75N fragment decreased the maximum tension by about 15% and increased the minimum tension by about 4-fold in the absence of calcium. The normalized *p*Ca-stiffness dependences of slow fibers with C0-C2 fragments are shown in Figure 1b. Table 2 shows the averaged values of the parameters of the calcium dependence of fiber stiffness: calcium sensitivity, *p*Ca_50_ (*p*Ca, at which the stiffness is half maximum), Hill coefficient, *h*, and the ratios of the maximum and minimum stiffness values *S*_maxC_/*S*_max_ and *S*_minC_/*S*_min_.

The calcium dependence of the fiber stiffness qualitatively repeated the dependence of tension. In contrast to tension, the maximum stiffness at the saturated calcium concentration dropped by about 18% upon the addition of any of the cMyBP-C fragments studied (Table 2). At the same time, the minimum stiffness with WT C0-C2 and P161S C0-C2 fragments did not change in the absence of calcium. The cMyBP-C fragment with the mutation D75N increased the minimum stiffness by approximately 2.6-fold.

To study the mechanical and kinetic characteristics of actin-myosin interactions in muscle fibers and the effects of the C0-C2 fragments, we used two mechanical protocols. One of them (*Protocol 1*) was a 10% fiber shortening and stretch to the initial length [23]. This protocol mimics the onset of contraction in permeabilized muscle fiber and allows for the general viewing of the fiber response. The typical mechanical response of fiber tension on the length perturbations and changes in sarcomere length are shown in Figure 2.

The rate of tension redevelopment in the fiber at the saturated calcium concentration was reduced by approximately 10% in the presence of any of the C0-C2 fragments compared to the rate in the control; the averaged *k*_trC_/*k*_tr_ ratio was 0.89 ± 0.03 (Figure 2, Table 3). At the low calcium level (*p*Ca 6.8), the rate of tension recovery in the fiber with WT C0-C2, on the contrary, increased by almost 1.5-fold regarding the rate in its presence (0.19 ± 0.03 s^−1^ vs. 0.34 ± 0.01 s^−1^; Figure 2). The rate of tension redevelopment of the fiber at the low calcium was fitted into a linear function due to its prolonged growth. The calcium dependence of the tension recovery rate of the fiber after detachment of cross-bridges correlated with the calcium curve of the fiber stiffness (Table 2, Figure 1b), i.e., it decreased at the saturated calcium concentration and increased at the low calcium levels. The same applied to each of the HCM fragments of cMyBPc in this study. At the intermediate calcium concentration (*p*Ca 6.5), the D75N C0-C2 fragment increased *k*_tr_ from 0.31 ± 0.02 s^−1^ to 2.66 ± 0.08 s^−1^, and the P161S fragment decreased it from 0.30 ± 0.02 s^−1^ to 0.22 ± 0.02 s^−1^, which corresponds to changes in the fiber stiffness at this calcium concentration. Therefore, the kinetics of tension redevelopment in the fiber was presumably determined by the number of cross-bridges.

However, as the tension transient with *Protocol 1* consists of two oppositely directed processes—relaxation due to the detachment of ‘old’ cross-bridges and tension rise because of the formation of ‘new’ ones—the correct kinetics of each process are masked. For this reason, we used another protocol (*Protocol 2*) [26], which was a 2% stretch of the fiber followed in 400 ms by a 2% release (Figure 3). Unlike *Protocol 1*, it allows a separate analysis of these two processes.

The rates of the force relaxation *k*_rel_ and development *k*_fd_ at the maximum Ca^2+^ activation fitted by single exponentials and their constants are shown in Table 3. The WT fragment C0-C2 did not affect any of the rates. The mutation D75N slowed the kinetics of force development by almost 1.2-fold but did not affect the relaxation rate. The mutation P161S slowed relaxation and force development by 1.3-fold and 1.25-fold, respectively. An example of the experimental recording of the stretch-release protocol (*Protocol 2*) in the presence of the P161S C0-C2 fragment is shown in Figure 3.

## 3. Discussion

In a previous work [21], we investigated the effects of the mutations D75N and P161S in cMyBP-C associated with hypertrophic cardiomyopathy on actin-myosin interactions at the molecular level; we found that they disrupt calcium regulation and, in addition, that the mutation D75N affects myosin function.

The mutation D75N was found in at least three individuals with hypertrophic cardiomyopathy [27,28], but functional studies for this variant have not been reported. In the ClinVar database [https://www.ncbi.nlm.nih.gov/clinvar/, accessed on 1 September 2024] the mutation D75N is classified as a Variant of Uncertain Significance. The mutation P161S was found sporadically in different human populations in patients with HCM [29,30,31,32], but the clinical description in the case of this mutation has not been described in the literature. In the ClinVar database, the mutation P161S is classified as pathogenic/uncertain significance. Here, we studied the effect of the mutations D75N and P161S on the mechanical and kinetic characteristics of muscle contraction using isolated permeabilized fibers from the rabbit soleus muscle. Slow skeletal muscle fibers were used as a model of cardiac cells since the heavy chains of slow skeletal and ventricular myosin are similar. The calcium dependence of the fiber tension and stiffness and the kinetics of tension recovery induced by a 10% release and stretch of the fiber were investigated. The results of studies at the molecular and cellular levels allow us to make assumptions about the possible mechanisms of myocardial contractility impairment in these cMyBP-C mutations.

### 3.1. Effects of the Mutations on Skeletal Muscle Fiber Contractile Properties

We obtained interesting results on the effect of D75N and P161S C0-C2 on force generation at non-saturating calcium concentrations and force development. The WT C0-C2 fragment increased force production at non-saturating calcium concentrations and force redevelopment similarly to that in isolated myocardial strips [14,15]. Fragments with mutations behaved differently. At non-saturating calcium concentrations, the D75N C0-C2 fragment increased tension and the rate of tension redevelopment compared to the WT fragment, whereas P161S C0-C2 decreased these characteristics (Table 1, Figure 1a). With the D75N fragment, the minimum tension increased 4-fold regarding the control at *p*Ca 9, whereas the minimum stiffness increased only 2.5-fold. Notably, in the presence of the C0-C2 fragment, the force per myosin molecule increased.

Lin and colleagues [33] showed that the N-terminal fragments of all MyBP-C isoforms—slow and fast skeletal and cardiac—can activate the cardiac thin filament, and the cardiac fragment does this most effectively. We also found that the WT C0-C2 fragment activates the thin filament, increasing force and the rate of force development at non-saturating calcium concentrations in both slow muscle fiber and cardiac proteins [21]. However, the effect of the mutation D75N in these two cases was different. In the motility assay, the mutation D75N decreased the Ca^2+^ sensitivity of thin filament sliding, and in the muscle fibers, it increased the Ca^2+^ sensitivity of the fiber tension to a greater extent than the WT C0-C2 fragment (Figure 1a, Table 1). There are several explanations for the difference in these results. First, this may occur due to peculiarities of the isoform composition of cardiac and slow skeletal muscles. Although myosin heavy chains of slow skeletal and cardiac muscles are identical, other proteins, like myosin light chains, particularly the regulatory ones, and tropomyosin differ. Slow muscles contain gamma tropomyosin, and the activation of thin filaments containing gamma tropomyosin differs from the activation of filaments with alpha tropomyosin [34]. Secondly, unlike the ensemble of molecules in the in vitro assay, the muscle fiber is a well-structured system. The C0-C2 fragment can interact with both myosin and the thin filament. The C0-C2 fragment can interact with both myosin and the thin filament. It is possible that the mutation D75N in the C0 domain, which binds to myosin RLC [35], affects the distance of myosin heads to the thin filament, promoting its activation. In addition, the difference in the effects may result from different experimental conditions: isometric contraction of the fiber vs. sliding in the motility assay.

In contrast to the D75N C0-C2 fragment, the P161S C0-C2 fragment significantly reduced the Ca^2+^ sensitivity of both the tension and the stiffness compared to the WT C0-C2 fragment (Figure 1, Table 1 and Table 2). The mutation P161S is in the C1 domain of cMyBP-C, which contains binding sites for actin, myosin S2 [12], and tropomyosin [11,36]. The C1 domain was shown to interact with tropomyosin, thus involving cMyBP-C in the movement of the tropomyosin strand from the closed to the open position [11,36]. A decrease in tension and the number of cross-bridges with the P161S C0-C2 fragment indicate a deterioration in thin filament activation. In addition, we recently showed that the presence of the mutation P161S impairs the thin filament activation [21], thereby preventing the transition of tropomyosin from the closed to the open state and the cross-bridges formation.

The calcium dependence of the fiber stiffness qualitatively repeated the calcium dependence of tension (Figure 1; Table 1 and Table 2). The addition of 5 μM WT or P161S C0-C2 did not affect the maximum tension of the fiber at the saturated calcium concentration; only with the D75N C0-C2 cMyBP-C fragment the maximum tension reduced by about 15% compared to the control (Table 1). At the same time, the maximum stiffness with all the studied cMyBP-C fragments decreased by approximately 20% of the control (Table 2). It is worth noting that with the WT and P161S C0-C2 fragments, a decrease in the maximum stiffness (i.e., in the number of cross-bridges) (Table 2) did not lead to a reduction in the maximum tension (Table 1). Since both force-generating and non-force-generating cross-bridges contribute about equally to the fiber stiffness [37], the results indicate that in the presence of WT or P161S C0-C2 fragments, the relative fraction of force-generating cross-bridges becomes higher leading to an increase in the average force per a myosin head. In the case of the D75N C0-C2 fragment, the maximum tension and stiffness decreased proportionally, i.e., the tension decrease occurred because of a reduction in the number of cross-bridges.

### 3.2. Comparison of the Effects of the Mutations D75N and P161S on Contractile Properties with the Effects of Other Mutations

Currently, there are only a few papers devoted to the study of the effects of HCM-associated mutations in the N-terminal domains of the cMyBP-C molecule on the mechanical properties of cardiac muscle preparations. The studied cMyBP-C mutations had a different impact on the cross-bridge kinetics and calcium sensitivity of muscle preparations.

The E258K mutation (pathogenic/likely pathogenic [the ClinVar database]), located in the C1 domain of cMyBP-C, accelerated the formation and dissociation of cross-bridges in cardiac myofibrils, thereby speeding up the tension rise and relaxation phases at saturated calcium concentrations [17,38]. The Y235S (residue 237 in human sequence) mutation (pathogenic [the ClinVar database]) in the C1 domain of cMyBP-C also demonstrated increased cross-bridge kinetics in multicellular permeabilized myocardial preparations at the saturated calcium concentration [18]. The HCM-related E330K (residue 334 in human sequence) mutation (Conflicting classifications of pathogenicity [the ClinVar database]) in the cMyBP-C *m*-motif slowed the kinetics of cross-bridge formation at saturated calcium [19]. In our experiments, the mutations D75N and P161S did not affect the rate of tension development *k*_tr_ at saturated calcium concentration (Table 3, *Protocol 1*). At non-saturating calcium levels, mutation D75N increased the rate of tension rise, and mutation P161S decreased it since the calcium dependence on the rate of force redevelopment followed the calcium dependence on stiffness (Figure 1b). Both the mutations D75N and P161S slowed the rate of cross-bridge formation *k*_fd_; the mutation P161S also slowed the relaxation rate *k*_rel_, i.e., cross-bridge detachment (Table 3, *Protocol 2*).

The E330K mutation suppressed the calcium dependence of force in multicellular cardiac preparations, and the L348P mutation (corresponds to residue 352 in human sequence; Uncertain significance [the ClinVar database]), on the contrary, enhanced it [19]. In concentrations of 5 μM, the cMyBP-C fragment with the L348P mutation significantly increased the active force in the absence of calcium; at a saturated calcium concentration, it lowered the maximum force [20]. A similar effect was observed with the D75N cMyBP-C fragment at the same concentration (Figure 1a). Additionally, the mutation D75N increased calcium sensitivity. The effect of the P161S cMyBP-C fragment on the mechanical characteristics was similar to that of the E330K mutation, which also reduced the calcium sensitivity of force and did not affect the level of active force without calcium and the maximum force at saturated calcium [19].

The L348P mutation in transgenic mice increased end-systolic time and slowed the relaxation rate, and the E330K mutation had the opposite effect at the organ level and significantly shortened ejection duration. In transgenic mouse models, there was no overt systolic dysfunction, but in L348P-Tg, diastolic dysfunction appeared by markedly slowing and lengthening relaxation [19]. Since the effects of the mutations D75N and P161S on single-cell mechanical properties are similar to those of the L348P and E330K mutations, similar effects at the whole-heart level could be expected.

The molecular mechanisms of the effects of these mutations differ. The mutation E258K reduces contractile force and accelerates twitch kinetics by disrupting the interaction of cMyBP-C with S2 [17]. The mutations E334K and L352P are in a trihelix bundle (the THB region) in the C-terminal region of the M-domain. The mutation E334K in the C0-C2 fragment did not affect actin binding [39,40], and E330K reduced it [20] and enhanced myosin binding [39]. The mutation L352P increased both actin and myosin binding [39]. These changes in the interaction may affect cross-bridge kinetics [39] and thus contribute to HCM pathogenesis.

Experiments on muscle fibers showed that both mutations studied disrupt the calcium regulation of contractile force and affect the kinetics of its development. We found that mutation D75N increased the calcium sensitivity of the force, and mutation P161S decreased it. The mutation D75N suppressed the maximal isometric tension and increased the tension and stiffness at low calcium. The mutation D75N slowed the kinetics of the force recovery but did not affect the relaxation rate. The mutation P161S slowed both the relaxation and force recovery. This influence on the kinetics may impair cardiac diastolic function and cause myocardial hypertrophy. Previously, we showed that the mutations D75N and P161S dramatically corrupt the structure of the N-terminal domains [21]. In the in vitro motility assay, we found that both mutations disturb the thin filament activation and the mutation D75N impairs actin binding [21]. Altogether, the results obtained in the in vitro assay with ADP, molecular dynamics modeling, and muscle fiber experiments indicate that the mutation D75N affects myosin function. Thus, at the molecular and cellular level, we have established that the mutations D75N and P161S disrupt actin-myosin interactions and may be the reason for the HCM development.

## 4. Materials and Methods

Manipulations with experimental animals complied with the Directive 2010/63/EU of the European Parliament and the Council of the EU and were approved by the Ethics Committee of the Institute of Immunology and Physiology of the Ural Branch of the Russian Academy of Sciences (Protocol No. 05/20 dated 23 September 2020).

The studies were performed on skinned fibers of slow skeletal muscles (*m. soleus*) from a New Zealand rabbit. The procedure for permeabilizing muscle fibers was described previously [37]. Briefly, fiber bundles (1 to 2 mm in diameter, ~3 cm long) were tied to wooden sticks and cut loose from the soleus muscles. Each bundle was put into a plastic tube filled with the storage solution (MgATP 5 mM, EGTA 5 mM, MOPS 100 mM, K propionate 70 mM, DTT 1 mM, PMSF 10 μM, pH 7.0, glycerol 50% *v*/*v*) and oscillated on a shaker for 24 h at 4 °C. Then the bundles were transferred to a fresh storage solution and stored at –20 °C for several months.

Single fibers from 1.5 to 3 mm long were attached to a linear motor and a force transducer and placed in an experimental cell with a relaxing solution (MOPS 100 mM, MgATP 5 mM, MgCl_2_ 6.5 mM, EGTA 5 mM, DTT 1 mM) at 4 °C. During the experiment, the fiber was continuously illuminated by a beam of 2 mW LED laser (wavelength at 635 nm, 1 mm in diameter), and the sarcomere length was monitored by the position of the first order diffraction maximum on a position-sensitive photodiode (LSC/30D; United Detector Technology, Hawthorne, CA, USA). The sarcomere length in the relaxed state was set to 2.4 μm.

The calcium dependencies of fiber tension and stiffness were recorded at 15 °C. Free calcium concentration was set using the Maxchelator program [https://somapp.ucdmc.ucdavis.edu/pharmacology/bers/maxchelator/, accessed on 15 January 2022]. The fiber was transferred into an activating solution for 5 s, and then the experimental protocol was performed.

N-terminal fragments C0-C2 of human recombinant cMyBP-C expressed in *E. coli* [21] were used. The studied C0-C2 fragments were added to the relaxing solution in a final concentration of 5 μM, and the fiber was incubated for 30 min at 4 °C. After that, the temperature in the cell was raised to 15 °C, and the calcium dependence of tension and stiffness were recorded. Using Western blot and confocal microscopy, Kunst et al. [14] showed that the cMyBP-C fragments penetrate the skinned skeletal muscle fibers and are homogeneously distributed in the sarcomere.

The fiber stiffness was measured by changes in tension in response to 0.2% amplitude sinusoidal oscillations of the fiber length at 1 kHz [37]. Calcium dependencies of the fiber tension and stiffness were approximated with the Hill equation:(1)$$Q=\frac{Q_{max}}{\left(1+10\cdot e^{h\cdot (pCa-{pCa}_{50})}\right)}$$
where *Q* and *Q*_max_ are tension *T* and the maximum tension *T*_max_ or stiffness *S* and the maximum stiffness *S*_max_ at the saturated calcium concentration, *h* is the Hill coefficient, and *p*Ca_50_ is calcium sensitivity (*p*Ca, at which *T* or *S* is half maximum).

All rate constants, *k*_tr_ (Figure 4), *k*_rel_, and *k*_fd_, are indexes of single exponential fitting of corresponding transients (Figure 3).

In all experiments, eight fibers from two rabbits (four fibers from each rabbit) were used. Differences in parameters were assessed by the nonparametric Mann–Whitney test at the significance level of *p* < 0.05. Data and statistical analysis were performed using the Origin 8 software (Origin Lab, Northampton, MA, USA).

## 5. Conclusions

cMyBP-C is a multidomain protein having multiple interactions with major sarcomere proteins. The mutations D75N and P161S, located in the C0 and C1 domains, respectively, affect the mechanisms of the calcium regulation of contraction, kinetics, and magnitude of muscle force. Each mutation is distinguished by a unique set of effects at the molecular and cellular levels of muscle organization, leading to contractility impairment. Our results, obtained with the mutation D75N, indicate the functional significance of the C0 domain.

## Figures and Tables

**Figure 1 ijms-25-13405-f001:**
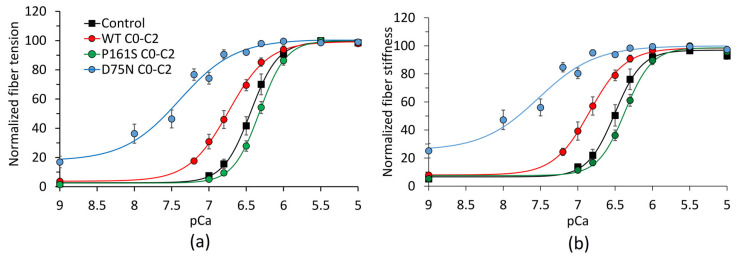
Normalized calcium dependences of slow muscle fiber tension (**a**) and stiffness (**b**) with WT, P161S, and D75N fragments of cMyBP-C. Symbols show experimental points, and lines are their approximation by the Hill Equation (1). Experimental points are shown as mean ± SD.

**Figure 2 ijms-25-13405-f002:**
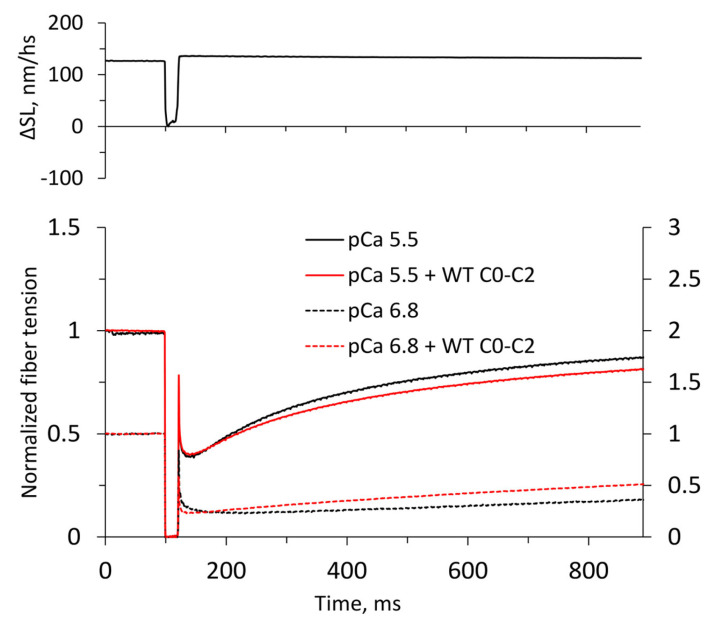
Tension redevelopment (lower panel) in muscle fiber after a run of *Protocol 1* at *p*Ca 5.5 (solid lines) and *p*Ca 6.8 (dashed lines) in the control and in the presence of the WT C0-C2 fragment of cMyBP-C. The left tension scale relates to *p*Ca 5.5, and the right one is for *p*Ca 6.8. The upper panel shows changes in the sarcomere length ΔSL in nm per half-sarcomere.

**Figure 3 ijms-25-13405-f003:**
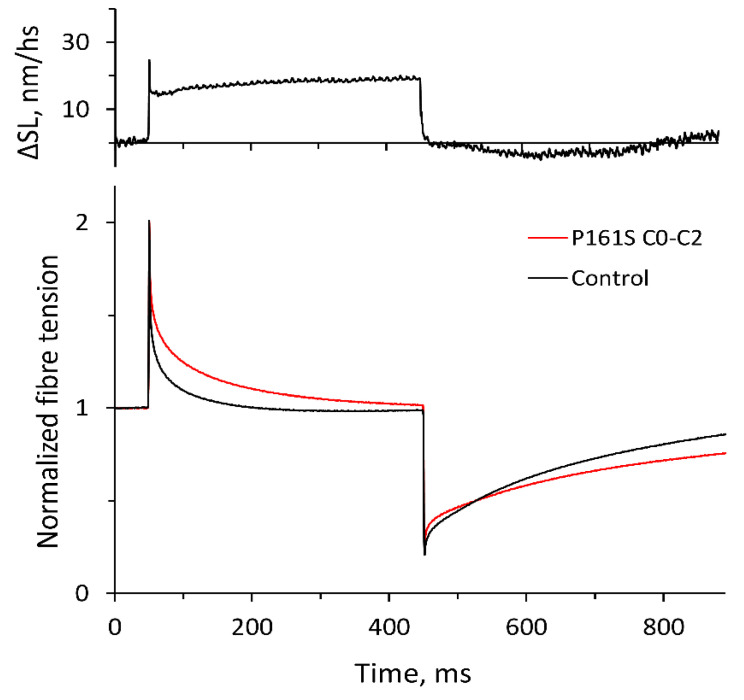
Example of the tension transient in the muscle fiber on the run of *Protocol 2* in the control (black line) and in the presence of the P161S C0-C2 fragment (red line). The upper panel shows changes in sarcomere length in nm per half-sarcomere.

**Figure 4 ijms-25-13405-f004:**
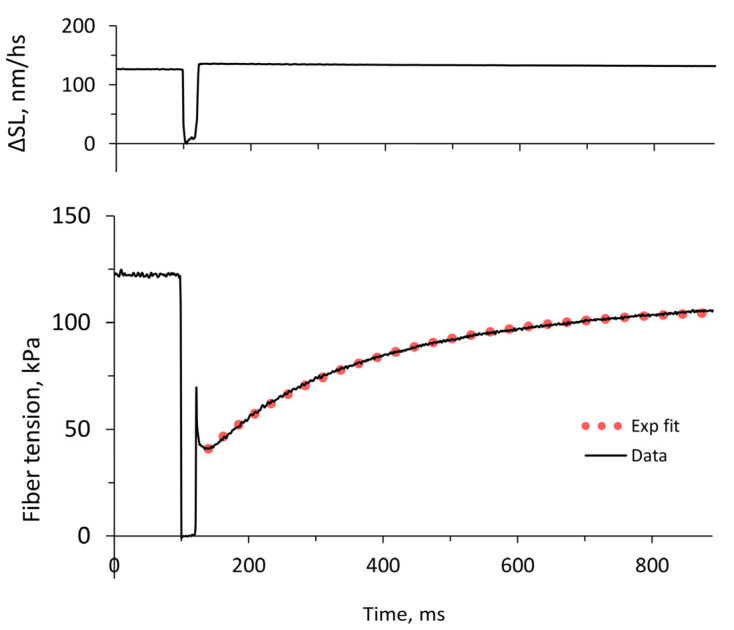
Typical tension response of a slow fiber to a release-stretch protocol at the saturating calcium concentration (*p*Ca 5.5). Changes in sarcomere length ΔSL in nm per half-sarcomere are shown in the upper panel. The tension response is fitted by an exponential function (red dots).

**Table 1 ijms-25-13405-t001:** Parameters of calcium dependence of tension in muscle fibers with cMyBP-C fragments.

Object	*p*Ca_50_	*h*	*T*_maxC_/*T*_max_ (*p*Ca 5.5)	*T*_minC_/*T*_min_ (*p*Ca 9)
Control	6.43 ± 0.04	2.47 ± 0.15	1	1
WT C0-C2	6.73 ± 0.04 *	1.59 ± 0.08 *	0.95 ± 0.01	1.38 ± 0.27
P161S C0-C2	6.31 ± 0.02 *^#^	2.46 ± 0.15 ^#^	0.96 ± 0.01	0.82 ± 0.14
D75N C0-C2	7.38 ± 0.07 *^#^	1.00 ± 0.11 *^#^	0.86 ± 0.02 *^#^	4.05 ± 0.36 *^#^

The symbol * indicates statistically significant differences in parameter values from those without cMyBP-C, and the symbol # indicates statistically significant differences in parameters compared to those with the WT C0-C2 fragment added. The statistical significance of the differences was assessed using the nonparametric Mann–Whitney test at a significance level of *p* < 0.05. The values are shown as mean ± SEM.

**Table 2 ijms-25-13405-t002:** Parameters of the calcium dependence of the muscle fiber stiffness with C0-C2 fragments of cMyBP-C.

Object	*p*Ca_50_	*h*	*S*_maxC_/*S*_max_ (*p*Ca 5.5)	*S*_minC_/*S*_min_ (*p*Ca 9)
Control	6.50 ± 0.04	2.59 ± 0.21	1	1
WT C0-C2	6.84 ± 0.05 *	1.96 ± 0.24 *	0.81 ± 0.01 *	1.00 ± 0.09
P161S C0-C2	6.37 ± 0.02 *^#^	2.50 ± 0.14 ^#^	0.82 ± 0.01 *	0.83 ± 0.07
D75N C0-C2	7.60 ± 0.16 *^#^	0.98 ± 0.16 *^#^	0.83 ± 0.02 *	2.55 ± 0.22 *^#^

The symbol * indicates statistically significant differences in the parameter values from those without cMyBP-C, and the symbol # indicates statistically significant differences in parameters compared to those with the WT C0-C2 fragment added. The statistical significance of the differences was assessed using the nonparametric Mann–Whitney test at a significance level of *p* < 0.05. The values are shown as mean ± SEM.

**Table 3 ijms-25-13405-t003:** The rate constants of the tension transients in the muscle fibers with cMyBP-C fragments obtained with the use of two mechanical protocols at *p*Ca 5.5.

Object	*Protocol 1*	*Protocol 2*			
	*K*_trC_/*k*_tr_	*k*_rel_, s^−1^		*k*_fd_, s^−1^	
		Control	with C0-C2	Control	with C0-C2
WT C0-C2	0.91 ± 0.03 *	12.2 ± 0.4	12.0 ± 0.3	4.5 ± 0.1	4.2 ± 0.1
D75N C0-C2	0.90 ± 0.01 *	12.7 ± 0.2	12.3 ± 0.2	4.6 ± 0.2	3.8 ± 0.1 *^#^
P161S C0-C2	0.87 ± 0.02 *	14.2 ± 0.4	11.1 ± 0.3 *^#^	4.9 ± 0.2	3.9 ± 0.1 *^#^

The symbol * indicates statistically significant differences in the parameter values from the control (i.e., without added cMyBP-C), and the symbol # represents significant differences compared to the WT C0-C2 fragment of cMyBP-C. The significance of the differences was assessed using the nonparametric Mann–Whitney test at a significance level of *p* < 0.05. The values are shown as mean ± SEM.

## Data Availability

The original contributions presented in this study are included in the article. Further inquiries can be directed to the corresponding author.

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
