# Peer review of "The N-Terminal Mutations of cMyBP-C Affect Calcium Regulation, Kinetics, and Force of Muscle Contraction"

_ijms, 2024, doi:10.3390/ijms252413405_

Round 1

Reviewer 1 Report

Comments and Suggestions for Authors

This study addressed the role of cMyBP-C N-terminal fragments on regulation of myofilament mechanics. The authors utilized rabbit soleus slow-twitch skeletal muscle fibers as an experimental model system to test cMyBP-C N-terminal fragments on isometric and dynamic contractile properties.  Results indicate that two different HCM cMyBP-C mutations uniquely alter calcium sensitivity of myofilament force and stiffness.  There are several concerns that should be addressed related to reliability of the data and appropriate interpretation.

  1.    Incorporation of cMyBP-C fragments. There lacks either quantitative or qualitative incorporation of cMyBP-C fragments into permeabilized slow-twitch skeletal muscle fibers.  Immunohistochemistry or gel electrophoresis are recommended to assess fragment incorporation and localization. This is necessary to assure group difference to do not arise from variable sarcomere incorporation. These measurements also would be helpful to explain whether fragments preferentially bind thin or thick filament proteins.

  2.    Sarcomere Length measurements. The authors report SL in nm/half sarcomere.  However, the Methods lack description on the system used to monitor sarcomere length.  If sarcomere length were directly measured, absolute sarcomere lengths should be reported.

  3.    Figure 2.  The force redevelopment traces are of low quality. There is high residual force after re-stretch and force seems to slowly dip before rising. In addition, force does not rise to near isometric values in during maximal Ca2+ activations as reported in other publications (J Physiol 501; 607-621, 1997; J Gen Physiol 151; 645-659, 2019). Please address, and provide absolute ktr values and discuss how they compare to previously reported values.

  4.    Figure 3.  Slow twitch skeletal muscle fibers often exhibit stretch activation (delayed transient force) after rapid stretch. Was stretch activation observed in these fibers?  If so, how does this affect the interpretation of the tension transients after stretch?

  5.    Table 3. Please clarify how krd relates to kfd, especially since these values often change in opposite direction.

 Mino

 1.    How were fiber bundles skinned?

 2.  Why was pCa 5.5 used for maximal Ca2+ activation?

Author Response

Reviewer 1

This study addressed the role of cMyBP-C N-terminal fragments on regulation of myofilament mechanics. The authors utilized rabbit soleus slow-twitch skeletal muscle fibers as an experimental model system to test cMyBP-C N-terminal fragments on isometric and dynamic contractile properties.  Results indicate that two different HCM cMyBP-C mutations uniquely alter calcium sensitivity of myofilament force and stiffness. There are several concerns that should be addressed related to reliability of the data and appropriate interpretation.

  1. Incorporation of cMyBP-C fragments. There lacks either quantitative or qualitative incorporation of cMyBP-C fragments into permeabilized slow-twitch skeletal muscle fibers. Immunohistochemistry or gel electrophoresis are recommended to assess fragment incorporation and localization. This is necessary to assure group difference to do not arise from variable sarcomere incorporation. These measurements also would be helpful to explain whether fragments preferentially bind thin or thick filament proteins.

To study the effects of cMyBP-C fragments on mechanical and kinetic characteristics of single permeabilized muscle fiber we used the method described by Kunst et al. (Kunst et al., Circ Res. 2000; Herron et al., Circ Res. 2006; Razumova et al., J Gen Physiol. 2008). With Western blot, they showed that the C0C1 and C1C2 fragments of cMyBP-C penetrate into skinned fibers and accumulate at appreciable levels. In addition, with confocal microscopy, the authors found that these fragments are homogeneously distributed in the sarcomere. Determining which sarcomere proteins the fragments colocalize with is an independent research issue requiring high-resolution fluorescence microscopy. We added information about the penetration of the cMyBP-C fragments in the skinned fibers to the text.

  1. Sarcomere Length measurements. The authors report SL in nm/half sarcomere. However, the Methods lack description on the system used to monitor sarcomere length. If sarcomere length were directly measured, absolute sarcomere lengths should be reported.

We added a description of sarcomere measurement procedure to the Methods section.

  1. Figure 2. The force redevelopment traces are of low quality. There is high residual force after re-stretch and force seems to slowly dip before rising. In addition, force does not rise to near isometric values in during maximal Ca2+activations as reported in other publications (J Physiol 501; 607-621, 1997; J Gen Physiol 151; 645-659, 2019). Please address, and provide absolute ktrvalues and discuss how they compare to previously reported values.

The residual force in rabbit slow fibers after the release-stretch protocol in our experiments averaged 38%. Previously published data (J Physiol 501; 607–621, 1997; J Gen Physiol 151; 645–659, 2019) show a residual force of 20–30% on either rabbit fast skeletal fibers or rat slow fibers. The slight excess of a residual force in our experiments can probably be due to species peculiarities since we used slow rabbit fibers. Quality of skinning is confirmed by the absence of tension and stiffness in the relaxing buffer (as seen from the calcium dependence of tension at pCa 9, Fig. 1a). Time resolution of our piezo crystal-based force transducer unfortunately, does not allow reliable recording protocols longer than 1 second, so the force in the fiber does not have time to recover completely during the protocol.

Following your remark, we changed krd to more traditional ktr. Measured ktr values ​​are given in the text, and the table shows the ratio of how much ktr changes in the presence of protein C fragments. A comparison of our ktr values ​​to previously published ones is added to the Discussion.

  1. Figure 3.  Slow twitch skeletal muscle fibers often exhibit stretch activation (delayed transient force) after rapid stretch. Was stretch activation observed in these fibers?  If so, how does this affect the interpretation of the tension transients after stretch?

Stretch activation of a muscle preparation, as shown, for example, in J Gen Physiol 151; 645–659, 2019, is observed on a time scale of tens of seconds. Due to the limited temporal resolution of the sensor we used, we cannot register such long-term changes.

  1. Table 3. Please clarify how krdrelates to kfd, especially since these values often change in opposite direction.

We replaced krd with ktr as suggested. The difference between ktr and kfd is described in the Results section, paragraph 2, p. 5. Virtually, these constants describe the same process, so their values ​​are close. These constants do not behave oppositely. They either do not change or decrease with the addition of protein-C fragments. The ktr values are in the range of earlier published data.

Mino

  1. How were fiber bundles skinned?

We added a description of skinning procedure to the Methods section.

  1. Why was pCa 5.5 used for maximal Ca2+activation?

We tested different pCa values ​​5, 5.5, 4. The maximum fiber tension was already achieved at pCa 5.5, so we limited the scale to this value. Replaced Figure 1 containing point pCa 5, shows this clearly.

Reviewer 2 Report

Comments and Suggestions for Authors

The authors are investigating the effects of the cMyBP-C mutations D75N and P161S in slow skeletal muscle fibers. It is well known that mutations to cMyBP-C can cause hypertrophic cardiomyopathy in humans. These mutations are in the N-terminal of cMyBP-C, which has not been as studied as other domains of the protein.

Though these mutations have been found in families, there is no direct correlation between the mutations and HCM mentioned in the manuscript. It is important for the authors to discuss the clinical significance of these particular mutations. How many people are affected by these mutations, and do they develop HCM? If so, how severe is the disease caused by these mutations?

In the discussion, the authors mention the mutations L352P and E334K, which had similar single-cell mechanical properties. In particular, L352P had diastolic dysfunction in transgenic mice, possibly predicting a possible whole heart result for D75N and P161S as well. The authors extrapolate from this data that these mutations may all have an outcome of HCM, though there is no clinical proof provided. Are the L352P and E334K mutations known to produce HCM in humans?

It is not certain that the information provided in the methods section is sufficient to allow for others to duplicate these experiments. For example, for the data in Figure 1, it is indicated in the results that eight fibers were used for each fragment. There is no indication of how many rabbits were used and how many fibers came from each animal. Figure 2 depicts a typical mechanical response, out of how many experiments? How many animals/fibers were used for Tables 2 and 3, and how many experiments does Figure 3 represent? Since the n is unclear, it is also unclear if the statistical analysis is adequate or correct.

The legend for Figure 1 indicates that the data is mean±SD, though Table 1 is mean±SEM. Tables 2 and 3 do not indicate if it is SD or SEM. This needs to be clarified and consistent.

Line 224. The sentence “Despite myosin heavy chains…” reads awkwardly.

Line 295. The sentence “Neither mouse model demonstrated…” reads awkwardly.

Author Response

Reviewer 2

The authors are investigating the effects of the cMyBP-C mutations D75N and P161S in slow skeletal muscle fibers. It is well known that mutations to cMyBP-C can cause hypertrophic cardiomyopathy in humans. These mutations are in the N-terminal of cMyBP-C, which has not been as studied as other domains of the protein.

Though these mutations have been found in families, there is no direct correlation between the mutations and HCM mentioned in the manuscript. It is important for the authors to discuss the clinical significance of these particular mutations. How many people are affected by these mutations, and do they develop HCM? If so, how severe is the disease caused by these mutations?

The D75N mutation was found in at least three individuals with HCM (Rodríguez-García et al., BMC Med Genet. 2010; Cecconi et al., Int J Mol Med. 2016; Robyns et al., Eur J Med Genet. 2020) but functional studies for this variant have not been reported. In the ClinVar database (https://www.ncbi.nlm.nih.gov/clinvar/variation/161313/), the D75N mutation is classified as a Variant of Uncertain Significance. The P161S mutation was found sporadically in different human populations in patients with HCM (Alders et al., Eur Heart J. 2003; Christiaans et al., Eur Heart J. 2010; Brito et al., Rev Port Cardiol. 2012; Verhagen et al., Eur J Hum Genet. 2018) but the clinical picture in case of this mutation has not been described in the literature. In the ClinVar database (https://www.ncbi.nlm.nih.gov/clinvar/variation/518242/), the P161S mutation is classified as Pathogenic/Uncertain Significance.

We added this information to the text.

In the discussion, the authors mention the mutations L352P and E334K, which had similar single-cell mechanical properties. In particular, L352P had diastolic dysfunction in transgenic mice, possibly predicting a possible whole heart result for D75N and P161S as well. The authors extrapolate from this data that these mutations may all have an outcome of HCM, though there is no clinical proof provided. Are the L352P and E334K mutations known to produce HCM in humans?

Yes, the L352P and E334K mutations were found in patients with HCM (Richard et al., Circulation, 2003). In the literature, there is no description of clinical findings for specific patients with these mutations, only general signs of HCM: maximal LV wall thickness >18 mm, fractional shortening 38.3±8.2%, systolic dysfunction etc. (Lopes et al., Heart. 2015;101(4):294-301. doi: 10.1136/heartjnl-2014-306387).

It is not certain that the information provided in the methods section is sufficient to allow for others to duplicate these experiments. For example, for the data in Figure 1, it is indicated in the results that eight fibers were used for each fragment. There is no indication of how many rabbits were used and how many fibers came from each animal. Figure 2 depicts a typical mechanical response, out of how many experiments? How many animals/fibers were used for Tables 2 and 3, and how many experiments does Figure 3 represent? Since the n is unclear, it is also unclear if the statistical analysis is adequate or correct.

In all experiments, eight fibers from two rabbits (four fibers from each) were used. We added this information to the Materials and Methods section.

The legend for Figure 1 indicates that the data is mean±SD, though Table 1 is mean±SEM. Tables 2 and 3 do not indicate if it is SD or SEM. This needs to be clarified and consistent.

The data on tension and stiffness are presented in Figure 1, and the characteristics of the Hill equation are collected in Table 1. For this reason, the data is presented differently. We indicated that the data in Tables 2 and 3 is mean±SEM.

Line 224. The sentence “Despite myosin heavy chains…” reads awkwardly.

We have rewritten this sentence.

Line 295. The sentence “Neither mouse model demonstrated…” reads awkwardly.

We have rewritten this sentence.

Round 2

Reviewer 1 Report

Comments and Suggestions for Authors

The authors adequately responded to previous comments.

Reviewer 2 Report

Comments and Suggestions for Authors

The authors have addressed all of my concerns.